

# A robust algorithmic cum integrated approach of interval-valued fuzzy hypersoft set and OOPCS for real estate pursuit

Muhammad Arshad[1], Muhammad Saeed[1], Atiqe Ur Rahman[1], Mazin Abed Mohammed[2], Karrar Hameed Abdulkareem[3], Abed Saif Alghawli[4] and Mohammed A.A. Al-qaness[5]

[1] Department of Mathematics, University of Management and Technology, Lahore, Pakistan
[2] College of Computer Science and Information Technology, University of Anbar, Anbar, Ramadi, Iraq
[3] College of Agriculture, Al-Muthanna University, Samawah, Iraq
[4] Computer Science Department, College of Sciences and Humanities, Prince Sattam bin Abdulaziz University, Al-Kharj, Saudi Arabia
[5] College of Physics and Electronic Information Engineering, Zhejiang Normal University, Jinhua, China

Corresponding author
Abed Saif Alghawli,
a.alghauly@psau.edu.sa

## ABSTRACT

Due to the vast variety of aspects that must be made—many of which are in opposition to one another—choosing a home can be difficult for those without much experience. Individuals need to spend more time making decisions because they are difficult, which results in making poor choices. To overcome residence selection issues, a computational approach is necessary. Unaccustomed people can use decision support systems to help them make decisions of expert quality. The current article explains the empirical procedure in that field in order to construct decision-support system for selecting a residence. The main goal of this study is to build a weighted product mechanism-based decision-support system for residential preference. The said house short-listing estimation is based on several key requirements derived from the interaction between the researchers and experts. The results of the information processing show that the normalized product strategy can rank the available alternatives to help individuals choose the best option. The interval valued fuzzy hypersoft set (IVFHS-set) is a broader variant of the fuzzy soft set that resolves the constraints of the fuzzy soft set from the perspective of the utilization of the multi-argument approximation operator. This operator maps sub-parametric tuples into a power set of universe. It emphasizes the segmentation of every attribute into a disjoint attribute valued set. These characteristics make it a whole new mathematical tool for handling problems involving uncertainties. This makes the decision-making process more effective and efficient. Furthermore, the traditional TOPSIS technique as a multi-criteria decision-making strategy is discussed in a concise manner. A new decision-making strategy, "OOPCS" is constructed with modifications in TOPSIS for fuzzy hypersoft set in interval settings. The proposed strategy is applied to a real-world multi-criteria decision-making scenario for ranking the alternatives to check and demonstrate their efficiency and effectiveness.

# INTRODUCTION

Many people find it challenging to select a residence in which to live (*Supriyono & Sari, 2018*). Cost, covered area, land size, the material used in architecture, number of bathrooms and bedrooms, green surroundings, access to main roads, distance to work, distance to a public park, distance to the main market, and so on are all components that individuals deem when purchasing a residence. Furthermore, certain of these standards are completely contradictory to one another, such as the price *versus* the material used in construction *versus* the size of the residence. The price of a house tends to increase with its size. The tendency among people is to have a large house with all possible basic facilities for the least amount of money. It is a task requiring decision-making (DM) using various attributes and sub-attributes considered at a time. However, since it requires expert knowledge to solve, the residence selection issue can be considered semi-structured (*Turban et al., 2007*). This is because it cannot be accomplished using general mathematical rules. To avoid any problems, people normally seek assistance from individuals who have purchased a residence before or from an advisor who serves as a professional. Such guidance is very informative, especially if it pertains to the skills and experience needed to assist in DM. Yet, according to *Badiru & Cheung (2002)*, there are numerous detriments to consulting a professional for selection, including the varied levels of competence or the unavailability of professionals, the unavailability owing to physical or emotional stress, the exclusion of essential components in a situation, unpredictable DM within the same context, the inability to retain and remember all relevant information or the difficulty in recalling or comprehending large amounts of data in a relatively short timeframe, and the inability to retrieve or interpret large data sets. Decision-making may be influenced by individual considerations, a lack of accountability after choices have been made, deception, and other variables that might reduce competence. Several initiatives have been undertaken by researchers to construct a decision support system enabling multi-criteria DM in a variety of contexts, including the selection of scholarship recipients (*Uyun & Riadi, 2011*).

Decision-making is the most prominent process that affects human behavior and occurs in a variety of contexts in the real world, including those related to the military, service, business, management, and other similar domains. The information required for making decisions, however, may not always be certain in actual situations. The process of DM starts with analyzing unclear information. Therefore, *Zadeh (1965)* presented the fuzzy set (F-set) theory to express fuzzy data mathematically. In such a set, each member of a specific set is characterized by a particular degree called the belonging degree, which is meant to measure its membership in that set. In other words, F-set is developed as a generalization of the phrase well-defined used in the definition of classical set. The F-set attracted the attention of several authors, but the recent researches (*Al-shami & Mhemdi, 2023*; *Al-shami, 2022*; *Rahman et al., 2020*) are worth noting regarding the introduction

of F-set variants and their utilization in DM. While dealing with a great deal of data, it is a very time-consuming activity to characterize the entities of such information one by one, thus F-set has limitations in such cases. Therefore, *Zadeh (1975a)*, *Zadeh (1975b)* and *Zadeh (1975c)* extended his own concept and initiated the concept of an interval-valued fuzzy set (IVF-set) which is mainly meant to characterize the entities present in a large amount of informational data. In this set, lower and upper bounds in terms of fuzzy values are used for the characterization of entities in the particular set. In this way, this set is more flexible as compared to F-set. Later on, it is observed that both F-set and IVF-set are not compatible with the parameterization scenario. Consequently, *Molodtsov (1999)* put forward the idea of the soft set (S-set) as a completely new parameterized class for estimating uncertainty that is free of this constraint. For the sake of the applicability of S-set in other fields of knowledge, the researchers (*Maji, Biswas & Roy, 2003*; *Ali et al., 2009*; *Çağman & Enginoğlu, 2010*; *Çağman, 2014*; *Zhu & Wen, 2013*) provided a number of soft set operations and their corresponding characteristics. By combining the F-set and IVF-set with S-set, *Maji, Biswas & Roy (2001)* and *Yang et al. (2009)* presented fuzzy soft set (FS-set) and interval-valued fuzzy soft set (IVFS-set), respectively. Recently, the authors (*Jan, Gwak & Pamucar, 2023*; *Al-shami, Alcantud & Mhemdi, 2023*; *Palanikumar & Iampan, 2022*) discussed the hybrids of F-set and IVF-set with modifications to their certain results.

In 2018, Smarandache observed that S-set is not compatible with those scenarios that enforce the classification of attributes into their relevant sub-attributive values in terms of nonoverlapping sets; therefore, he put forward a novel parameterized structure called the hypersoft set (*Smarandache, 2018*) (HS-set) which is capable of easing the decision makers burden by providing multi-argument approximations for the evaluation of alternatives. *Saeed et al. (2022)* discussed the various operations, matrix manipulation, and fundamental results of HS-set with numerical illustration. *Ihsan et al. (2022b)* and *Ihsan et al. (2022a)* put forward the ideas of a bijective hypersoft expert set and a hypersoft expert set, respectively, and applied them in DM scenarios. By combining F-set with HS-set, *Yolcu & Ozturk (2021)* proposed fuzzy hypersoft set (FHS-set) and discussed its application in DM scenario. Afterward, *Debnath (2021)* formulated a weighted operator of the FHS-set and applied it in DM scenario. *Ihsan, Rahman & Saeed (2021)* and *Kamacı & Saqlain (2021)* proposed the idea of a fuzzy hypersoft expert set and characterized its operations and properties. *Rahman, Saeed & Smarandache (2022)* transformed the classical idea of convex and concave sets in FHS-set environment and modified their results. Recently, *Saeed et al. (2023)* proposed interval valued fuzzy hypersoft set (IVFHS-set) by combining IVF-set and HS-set. They explained several rudiments and provided detailed numerical illustrations. *Arshad et al. (2023)* and *Arshad et al. (2022)* applied the idea of IVFHS-set in recruitment-based pattern recognition and evaluation of prescription consequences in Omicron patients by using the formulation of similarity and distance measures.

Technique for Order Preference by Similarity to Ideal Solution (TOPSIS), created by *Hwang & Yoon (1981)*, is a multi-criteria DM (MCDM) method that is used to determine the best option from a set of alternatives. Fuzzy TOPSIS is an extension of the traditional TOPSIS method that takes into account uncertainty and vagueness in DM. It was first

proposed by *Chen (2000)*. In Fuzzy TOPSIS, the preferential values of substitutes with regard to criteria are described by fuzzy numbers instead of crisp numbers. Fuzzy numbers allow for the representation of imprecise and uncertain information in a decision problem. The weights of the criteria are also represented by fuzzy numbers. The steps involved in Fuzzy TOPSIS are similar to those of traditional TOPSIS, with modifications such as fuzzification of the decisive and fuzzy weighted normalized decisive matrix, determination of the fuzzy ideal and anti-ideal solutions, calculation of the distance to the fuzzy ideal and anti-ideal solutions, calculation the relative closeness to the fuzzy ideal solution, and ranking of the alternatives. Fuzzy TOPSIS is useful for DM when there is uncertainty and vagueness in the decision problem. It allows decision-makers to consider imprecise and uncertain information in a systematic and objective way. However, like all DM methods, it has its limitations and assumptions, and its results should be interpreted with caution.

## Motivation of proposed study

Many researchers made rich contributions regarding the utilization of fuzzy TOPSIS for handling various DM situations. *Eraslan (2015)* presented a DM system structured on TOPSIS and soft set theory. *Eraslan & Karaaslan (2015)* introduced a DM TOPSIS technique based on a fuzzy soft environment. *Ashtiani et al. (2009)* and *Mokhtarian (2015)* extended TOPSIS method to interval-valued fuzzy sets. *Tripathy, Sooraj & Mohanty (2017)* employed a novel approach to IVFS-set for discussing DM situation. The HS-set provides a more simplistic modified version of the S-set that settles the barriers of the FS-set by making use of the multi-argument approximation operator (MAAO) instead of the single-argument approximation operator (SAAO). This tool maps sub-parametric entities into a power set of the universal set. It focuses on attribute segmentation into a non-overlapping attribute value set. These features make it a completely new mathematical tool for dealing with situations involving uncertainty and risk. This enhances the effectiveness and efficiency of DM. In some cases, DM requires a more strategic approach than just selecting the best available goods or services. In such situations, what is paramount may rely on numerous factors. One such case may occur when the experts are hesitant and they give their opinion in terms of linguistic values that are required to be transformed to interval-valued fuzzy values, *i.e.,* membership degree for approximating an alternative based on opted attributes to deal with roughness-based imprecision.

As the TOPSIS approach heavily relies on multi-parameter DM, a sophisticated MADM has been employed to rate potential residential construction options. In order to do this, a brand-new approach called "OOPCS (optimal order preference correlation strategy)" is created on the foundation of FHS-sets with interval settings. By using this technique, a collection of characteristics can be divided into several sub-attributed valued sets, where each attribute correlates to a different valued set. Data might be in the range between the lower bond and the higher bond due to the intervals employed in this approach. Decision makers assign weight in two phases. In the first phase, decision-makers rank each component of the HS-set without taking into account the alternatives. In order to avoid bias, each k-tuple element of the hypersoft model is assigned a weight based on the preferences expressed by each decision-maker. A weighted vector is then created that

maintains the relevance of each tuple member at a constant level. In the second phase, each decision-maker ranks each alternative based on its associated tuple value. This dual ranking minimizes the bias-ness of decision-makers towards any specific alternative. After observing the above literature, it is quite transparent that there is a need to initiate a mathematical framework that may tackle the following concerns and issues collectively:

1. How can the limitations of IVFS-set like structures be managed regarding the partitioning of attributes into their related non-intersecting subclasses consisting of subattributive values?

2. How can traditional TOPSIS be modified for a multi-argument approximate operator?

The proposed mathematical structure, *i.e.,* IVFHS-set, can easily tackle the above issues through their integration and characterization.

The novelty of the proposed framework is that the adopted mathematical structure, *i.e.,* IVFHS-set, is more flexible as compared to relevant existing literature because it generalizes most of the pre-developed fuzzy and soft set like structures. It not only copes with a large amount of data by introducing lower and upper bounds but also assists the decision-makers in making decisive comments by considering multiple arguments simultaneously. The proposed framework is actually an integration of IVFHS-set theory and its modified TOPSIS which has not been addressed by anyone so far in the literature.

The significant contributions of the study are outlined below:

1. The well-known DM technique TOPSIS is modified for IVFHS-sets by considering multi-argument approximate mapping with fuzzy graded approximations.

2. An innovative DM strategy called OOPCS is established, which employs a modified TOPSIS method and weight vectors of parametric tuples-based matrices and alternative-based matrices for the evaluation process by considering related decision makers' approximations.

3. A robust algorithmic approach is utilized to evaluate appropriate residential buildings by integrating modified TOPSIS, OOPCS, and aggregation operations of the IVFHS-set. Moreover, the flexibility and reliability of the proposed strategy are assessed through structural comparison.

This research work is structured as follows: Section 2 outlines the fundamentals of IVFHS-sets, F-sets, and S-sets, along with their interval-valued hybrids. Section 3 describes the main process for the standard HS in a sequence of steps. In Section 4, a new strategy TOPSIS is developed based on OOPCS-set theory for group IVFHS mechanisms. Section 5 demonstrates the efficiency of the proposed method through a real-world application along with a comparative study and sensitivity analysis. Section 6 concludes the research work with future directions.

## PRELIMINARIES

This portion of article demonstrates the basic notions from the literature, especially from *Gorzałczany (1987)*, *Molodtsov (1999)*, *Smarandache (2018)* and *Saeed et al. (2023)*.

**Definition 2.1** (*Gorzałczany, 1987*) The set of entities $(\ddot{\theta}, \zeta_F(\ddot{\theta}))$ is claim to be an IVF-set on $\ddot{\theta}$ (initial space of objects) when for any $\ddot{\theta} \in \ddot{\theta}$, $\zeta_F : \ddot{\theta} \to C(\ddot{I})$ such that the value $\zeta_F(\ddot{\theta})$ is

characterized in terms of closed interval. It is meant to elaborate the bounds for belonging grades of $\ddot{\theta} \in \ddot{\theta}$. The family of IVF-sets is symbolized as $C(IVFS)$.

For the sake of getting multi-argument domain, *Smarandache (2018)* initiated the idea of HS-set that is, in fact, an extension of S-set (*Molodtsov, 1999*).

**Definition 2.2** (*Smarandache, 2018*) The set of approximate elements $\zeta_H(\acute{\epsilon}_i)$ characterized by approximate mapping $\zeta_H : \ddot{\Lambda} \rightarrow 2^{\ddot{\theta}}$, is claimed to be HS-set on $\ddot{\theta}$ such that $\ddot{\Lambda}$ is the product of non overlapping attributive subclasses with regards to different attributes $\ddot{\eth}_i$.

Recently *Saeed et al. (2023)* studied the various axiomatic properties and operations of IVFHS-sets by combining the ideas of IVF-sets and HS-sets.

**Definition 2.3** (*Saeed et al., 2023*) The set of approximate elements $\zeta_F(\ddot{\epsilon}_i)$ characterized by approximate mapping $\zeta_F : \ddot{\Lambda} \rightarrow C(IVFS)$, is claimed to be IVFHS-set on $\ddot{\Theta}$ such that $\ddot{\Lambda}$ is treated as the same as defined in Definition 2.2.

# METHODOLOGY OF THE STUDY

Since multi-attribute DM is an important part of the TOPSIS technique, an intelligent MPDM has been used for ranking alternatives for the selection of residential buildings. For this purpose, a new strategy named ''OOPCS'' is developed on the basis of FHS-sets with interval settings. This strategy allows the partitioning of a set of attributes into disjoint sub-attributed valued sets in which each attribute corresponds to a unique valued set. The intervals used for this methodology allow data to be in the range between the lower bound and the upper bound. Initially, decision-makers rank each element of the HS-set. So each k-tuple element of the HS-set is given weight according to the preference given by each decision maker, and a weighted vector is constructed that keeps the importance of each tuple element constant so that bias might be reduced. Whereas in step 6, each decision maker ranks each alternative, so that each and every minute factor is taken into consideration. Another important factor of this strategy, ''OOPCS'' is the dual ranking of decision-makers: first, the ranking for the elements of HS-set tuples without the consideration of alternatives, and second, the ranking of alternatives on the basis of the corresponding tuple value. By increasing the number of decision-makers, one also reduces the factor of favoritism. Different steps of TOPSIS have been modified in OOPCS for the HS-set with interval settings. The TOPSIS technique is elaborated, modified, and applied to real-world DM problems. An IVFHS-set-based optimized framework for residential building selection by MPDM is demonstrated in Fig. 1.

## Modification of TOPSIS

The TOPSIS is a helpful and pragmatic strategy for evaluating and choosing a variety of alternatives employing distance measures. TOPSIS operations include decision matrix normalization, distance measurements, and aggregation operations (*Shih, Shyur & Lee, 2007*). A literature review of research (*Hwang & Yoon, 1981*; *Yoon, 1987*) on the TOPSIS technique is recommended for a better understanding of the concept. The TOPSIS technique adopted by *Eraslan & Karaaslan (2015)* has been described in this section, and

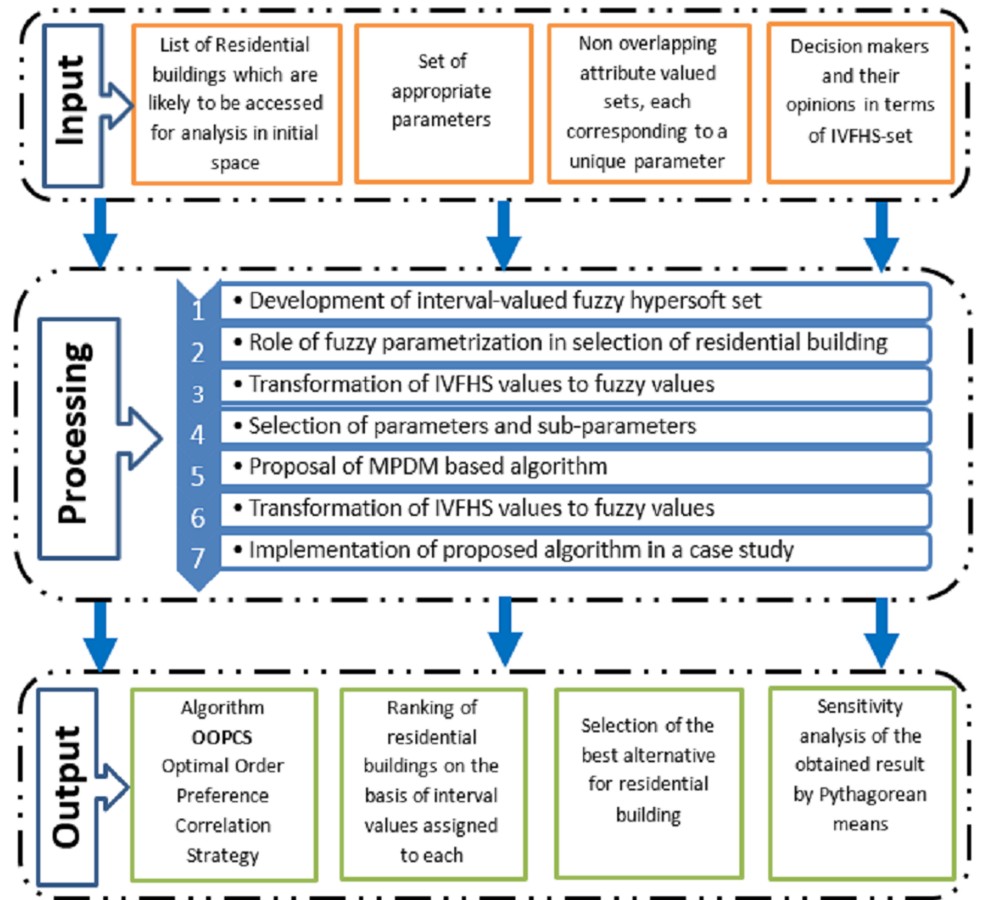

**Figure 1** An IVFHS-set based optimised framework for residential building selection by MPDM.

its modified form for the interval-valued hypersoft set is described in the next section. Step by step TOPSIS technique is described hereafter and demonstrated in Fig. 2.

Let $\aleph_n = \{1, 2, ..., n\} \forall n \in \mathfrak{N}$ (the set of natural numbers)

Step 1   Let $\mathbf{M_D}$ be the decision matrix defined in Eq. (1) demonstrated in the form of Table 1, where $\mathbf{A}_i, i \in \aleph$ represent alternative and $\zeta_j, j \in \aleph$ represent criteria.
$$\mathbf{M_D} = [\varphi_{ij}]_{m \times n} \tag{1}$$

Step 2   Construction of normalized decision matrix $\mathbf{D_N}$ displayed in Table 2.
Where each entry of normalized decision matrix $\mathbf{D_N}$ can be calculated using Eq. (2).
$$\gamma_{ij} = \frac{\varphi_{ij}}{\sqrt{\sum_{k=1}^{m} \varphi^2_{kj}}}, \forall \varphi_{ij} \neq 0 \tag{2}$$

Step 3   Formation of weighted normalized decision matrix $\mathbf{D_{WN}} = [\zeta_{ij}]_{m \times n} = [\varpi_j \gamma_{ij}]_{m \times n}, i \in \aleph_m$, displayed in the form of Table 3, where $\varpi_j = \frac{\mathbf{W}_j}{\sum_{k=1}^{m} \mathbf{W}_j}, j \in \{1, 2, ..., n\}$, such that $\sum_{k=1}^{m} \varpi_j = 1$, and $\mathbf{W}_j$ are weights for criteria $\zeta_j, j \in \aleph_n$.
$$\mathbf{D_{WN}} = [\zeta_{ij}]_{m \times n}$$

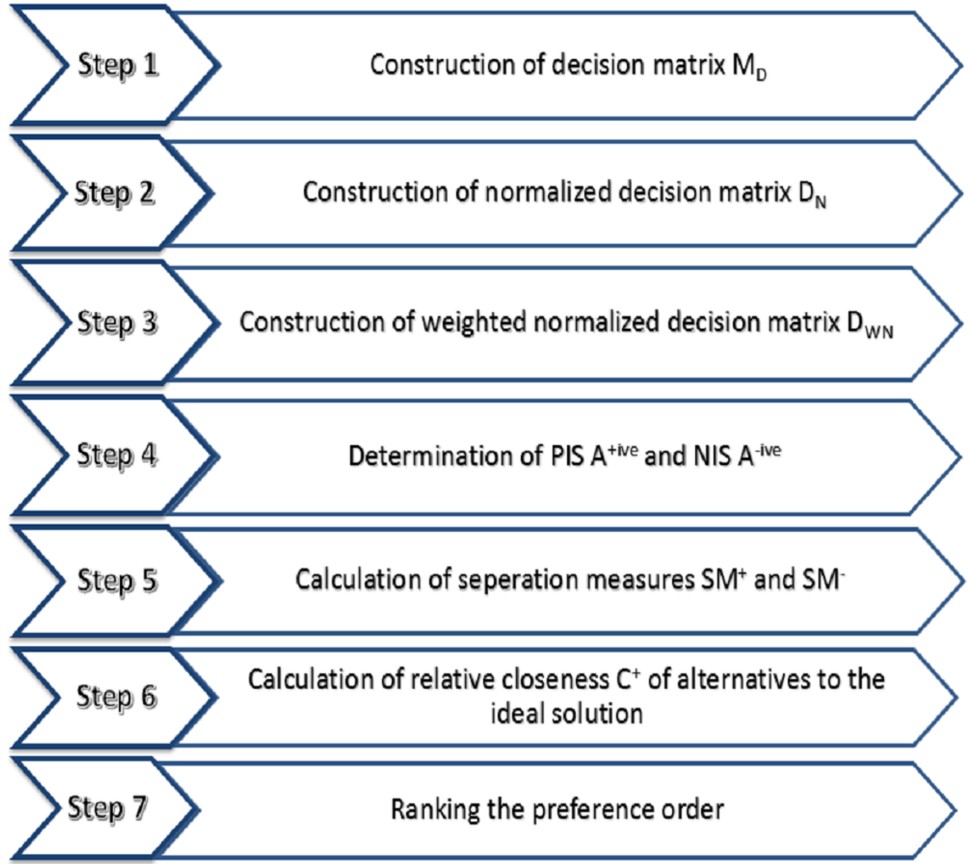

**Figure 2 Step by step TOPSIS technique.**

Step 4   Calculation of $\mathbf{A}^{+ive}$, the $+ive$ ideal solution (**PIS**) and $\mathbf{A}^{-ive}$, the $-ive$ ideal solution (**NIS**).

$$\mathbf{A}^{+ive} = \left\{\zeta_1^+, \zeta_2^+, ..., \zeta_j^+, ..., \zeta_n^+\right\} = \left\{\left(\max_i \zeta_{ij} | j \in \Upsilon_1\right), \left(\min_i \zeta_{ij} | j \in \Upsilon_2\right), i \in \aleph_m\right\} \qquad (3)$$

$$\mathbf{A}^{-ive} = \left\{\zeta_1^-, \zeta_2^-, ..., \zeta_j^-, ..., \zeta_n^-\right\} = \left\{\left(\min_i \zeta_{ij} | j \in \Upsilon_1\right), \left(\max_i \zeta_{ij} | j \in \Upsilon_2\right), i \in \aleph_m\right\} \qquad (4)$$

where $\Upsilon_1$ is benefit attribute set and $\Upsilon_2$ is cost attribute set.

Step 5   Calculation of separation measure of $+ive$ ideal $\mathbf{SM}_i^+$ solution and separation measure of $_ive$ ideal $\mathbf{SM}_i^-$ solution.

$$\mathbf{SM}_i^+ = \sqrt{\sum_{j=1}^{n} \left(\zeta_{ij} - \zeta_j^+\right)^2}, \forall i \in \aleph_m \qquad (5)$$

and

$$\mathbf{SM}_i^- = \sqrt{\sum_{j=1}^{n} \left(\zeta_{ij} - \zeta_j^-\right)^2}, \forall i \in \aleph_m \qquad (6)$$

**Table 1  Decision table $M_D$.**

| $M_D$ | Criteria $\longrightarrow$ | | | |
|---|---|---|---|---|
| **Alternatives $\downarrow$** | $\zeta_1$ | $\zeta_2$ | $\cdots$ | $\zeta_n$ |
| $A_1$ | $\varphi_{11}$ | $\varphi_{12}$ | $\cdots$ | $\varphi_{1n}$ |
| $A_2$ | $\varphi_{21}$ | $\varphi_{22}$ | $\cdots$ | $\varphi_{2n}$ |
| $\vdots$ | $\vdots$ | $\vdots$ | | $\vdots$ |
| $A_i$ | $\varphi_{i1}$ | $\varphi_{i2}$ | $\cdots$ | $\varphi_{in}$ |
| $\vdots$ | $\vdots$ | $\vdots$ | | $\vdots$ |
| $A_j$ | $\varphi_{j1}$ | $\varphi_{j2}$ | $\cdots$ | $\varphi_{jn}$ |

**Table 2  Normalized decision matrix $D_N$.**

| | | | |
|---|---|---|---|
| $\gamma_{11}$ | $\gamma_{12}$ | $\cdots$ | $\gamma_{1n}$ |
| $\gamma_{21}$ | $\gamma_{22}$ | $\cdots$ | $\gamma_{2n}$ |
| $\vdots$ | $\vdots$ | | $\vdots$ |
| $\gamma_{m1}$ | $\gamma_{m2}$ | $\cdots$ | $\gamma_{mn}$ |

**Table 3  Weighted normalized decision matrix $D_{WN}$.**

| | | | |
|---|---|---|---|
| $\zeta_{11}$ | $\zeta_{12}$ | $\cdots$ | $\zeta_{1n}$ |
| $\zeta_{21}$ | $\zeta_{22}$ | $\cdots$ | $\zeta_{2n}$ |
| $\vdots$ | $\vdots$ | | $\vdots$ |
| $\zeta_{m1}$ | $\zeta_{m2}$ | $\cdots$ | $\zeta_{mn}$ |

Step 6  Calculation of relative nearness of alternatives to the optimal solution

$$C_i^+ = \frac{SM_i^-}{(SM_i^- + SM_i^+)}, 0 \leq C_i^+ \leq 1, \forall i \in \aleph_m \tag{7}$$

Step 7  Ranking the preference order.

## OPTIMAL ORDER PREFERENCE CORRELATION STRATEGY "OOPCS" WITH INTERVAL-VALUED FUZZY HYPERSOFT SET INFORMATION FOR GROUP DM

A new strategy OOPCS is suggested in this section by enhancing TOPSIS (*Supriyono & Sari, 2018*; *Shih, Shyur & Lee, 2007*; *Hwang & Yoon, 1981*; *Yoon, 1987*; *Eraslan & Karaaslan, 2015*) to interval-valued fuzzy hypersoft set environment with modifications.

This method's core procedure illustrated with the help of Fig. 3 is described hereafter:

Step 1  Statement of the problem

Consider $\mathscr{D} = \{\Delta_1, \Delta_2, ..., \Delta_p\}$ as the set of decision makers, $\ddot{\Theta} = \{\ddot{\theta}_1, \ddot{\theta}_2, ..., \ddot{\theta}_m\}$ be the alternatives and $X = \{x_1, x_2, ..., x_n\}$ be the set of parameters. Then a hypersoft set can

**Figure 3  OOPCS strategy for group DM with interval-valued fuzzy hypersoft set information.**

be defined as

$$\xi : \mathfrak{X}^{\star} \to F(\ddot{\Theta}),$$

where $\mathfrak{X}^{\star} = X_1 \times X_2 \times ... \times X_n$ such that each attribute $x_1, x_2, ..., x_n$ corresponds to a unique disjoint attribute valued set $X_1, X_2, ..., X_n$. Let $\mho \in \mathfrak{X}^{\star}$.

Step 2  Construction of weighted interval valued fuzzy parameter hypersoft set represented in the form of matrix $\mathbf{M_D}$ displayed in Table 4.

$$\mathbf{M_D} = \left[ \varphi_{ij} \right]_{m \times n} \tag{8}$$

where $\varphi_{ij}$ is linguistic rating (see Table 5) assigned by decision maker $\Delta_i$, the sub-parametric tuple $\mho_j \in \mathfrak{X}^{\star}$.

Step 3  Calculation of mean difference of each interval of weighted interval-valued fuzzy parameter matrix $\mathbf{M_D}$ by $x_t = \frac{b_t - a_t}{2}$ obtained for each interval $(a_t, b_t) \in \mathbf{M_D}$

Step 4   Construction of weighted vector $\mathbf{W} = \{\mathbf{W}_1, \mathbf{W}_2, ..., \mathbf{W}_n\}$.
The elements of weighted vector $\mathbf{W}$ can be calculated by utilizing the Eq. (9)

$$\mathbf{W_j} = \frac{\varpi_j}{\sum_{k=1}^{m} \varpi_k}, \varpi_j = \frac{1}{m} \sum_{i=1}^{m} \varphi_{ij} \tag{9}$$

Step 5   Construction of fuzzy decision matrix $\mathbf{D}_k$ displayed in the form of Table 6 corresponding to each decision maker $\Delta_k$

Step 6   Construction of average interval-valued fuzzy hypersoft decision matrix $\mathbf{V}$ using Eq. (10)

$$\mathbf{V} = \frac{1}{n}(\mathbf{D}_1 \oplus \mathbf{D}_2 \oplus ... \oplus \mathbf{D}_n) = [\zeta_{ij}]_{m \times n} \tag{10}$$

$$[\zeta_{ij}]_{m \times n} = \left[ \frac{\zeta_{ij}^{l1} + \zeta_{ij}^{l2} + ... + \zeta_{ij}^{ln}}{n}, \frac{\zeta_{ij}^{u1} + \zeta_{ij}^{u2} + ... + \zeta_{ij}^{un}}{n} \right]_{m \times n}$$

where $\oplus$ represent matrices sum taken for corresponding lower bonds $\zeta^{l1}, \zeta^{l2}, ..., \zeta^{ln}$ and corresponding upper bonds $\zeta^{u1}, \zeta^{u2}, ..., \zeta^{un}$ of intervals $[\zeta^{l1}, \zeta^{u1}] \in \mathbf{D}_1^{\star}, [\zeta^{l2}, \zeta^{u2}] \in \mathbf{D}_2^{\star}, ..., [\zeta^{ln}, \zeta^{un}] \in \mathbf{D}_n^{\star}$ respectively:

Step 7   Construction of Mean difference of average interval-valued fuzzy parameter matrix $\mathbf{V}^{\star}$ by $x = \frac{b-a}{2}$ for interval $(a, b)$

Step 8   Construction of weighted fuzzy decision matrix $\mathbf{V_W}$ represented in the form of Table 7. where

$$\mathbf{V_W} = \mathbf{W}.\zeta_{ij} \tag{11}$$

Step 9   Figuring fuzzy-valued $+ive$ ideal solution $(^{+}IS)$ and fuzzy-valued $-ive$ ideal solution $(^{-}IS)$. $^{+}IS$ and $^{-}IS$ are obtained with the help of fuzzy set theory and the TOPSIS technique;

$$^{+}IS = \left\{ \breve{\xi}_1^{+}, \breve{\xi}_2^{+}, ..., \breve{\xi}_j^{+}, ..., \breve{\xi}_n^{+} \right\} = \left\{ \left( \max_j \zeta_{ij} | j \in \mathbb{J} \right), (\zeta_{ij} | j = 1, 2, ..., m), i \in \aleph_m \right\} \tag{12}$$

$$^{-}IS = \left\{ \breve{\xi}_1^{-}, \breve{\xi}_2^{-}, ..., \breve{\xi}_j^{-}, ..., \breve{\xi}_n^{-} \right\} = \left\{ \left( \min_j \zeta_{ij} | j \in \mathbb{J} \right), (\zeta_{ij} | j = 1, 2, ..., m), i \in \aleph_m \right\} \tag{13}$$

Step 10   Figuring separation measures $(\mathbf{SM}_i^{+})$ and $(\mathbf{SM}_i^{-})$ for each parameter by employing Eqs. (5) and (6).

Step 11   Calculation of the nearness of alternatives to the optimal solution $\mathbf{RC}_i^{+}$ using Eq. (7)

$$\mathbf{RC}_i^{+} = \frac{\mathbf{SM}_i^{-}}{\mathbf{SM}_i^{-} + \mathbf{SM}_i^{+}}, 0 \leq \mathbf{RC}_i^{+} \leq 1, \forall i \in \aleph_m$$

Step 12   Ranking the preference order.

## MPDM APPROACH FOR RANKING OF ALTERNATIVES BASED ON INTERVAL VALUED FUZZY HYPERSOFT SET USING OOPCS STRATEGY

A real-world scenario on fuzzy hypersoft set theory for a group DM method is discussed in this section. Table 8 summarizes the top attributes and their description together with their units.

### Strategy for selected parameters

The choice of residence is influenced by a number of parameters. Criteria (parameters) can be of decisive importance for the DM process. Therefore, careful selection and calculation

**Table 4   Decision table M$_D$.**

| M$_D$ | $\mho_1$ | $\mho_2$ | $\cdots$ | $\mho_n$ |
|---|---|---|---|---|
| $\Delta_1$ | $\varphi_{11}$ | $\varphi_{12}$ | $\cdots$ | $\varphi_{1n}$ |
| $\Delta_2$ | $\varphi_{21}$ | $\varphi_{22}$ | $\cdots$ | $\varphi_{2n}$ |
| $\vdots$ | $\vdots$ | $\vdots$ | | $\vdots$ |
| $\Delta_i$ | $\varphi_{i1}$ | $\varphi_{i2}$ | $\cdots$ | $\varphi_{in}$ |
| $\vdots$ | $\vdots$ | $\vdots$ | | $\vdots$ |
| $\Delta_m$ | $\varphi_{m1}$ | $\varphi_{m2}$ | $\cdots$ | $\varphi_{mn}$ |

**Table 5   Linguistic term for evaluation of parameters.**

| Linguistic term | IVF-value |
|---|---|
| Extremely important | $[0.86, 1.00]$ |
| Very important | $[0.66, 0.85]$ |
| Important | $[0.36, 0.65]$ |
| Un-important | $[0.16, 0.35]$ |
| Ir-relevant | $[0.00, 0.15]$ |

**Table 6   Fuzzy decision matrices D$_k$.**

| D$_k$ | $\mho_1$ | $\mho_2$ | $\cdots$ | $\mho_n$ |
|---|---|---|---|---|
| $\theta_1$ | $\varphi_{11}$ | $\varphi_{12}$ | $\cdots$ | $\varphi_{1n}$ |
| $\theta_2$ | $\varphi_{21}$ | $\varphi_{22}$ | $\cdots$ | $\varphi_{2n}$ |
| $\vdots$ | $\vdots$ | $\vdots$ | | $\vdots$ |
| $\theta_i$ | $\varphi_{i1}$ | $\varphi_{i2}$ | $\cdots$ | $\varphi_{in}$ |
| $\vdots$ | $\vdots$ | $\vdots$ | | $\vdots$ |
| $\theta_j$ | $\varphi_{j1}$ | $\varphi_{j2}$ | $\cdots$ | $\varphi_{jn}$ |

are required. It is a long-standing practice that residence is often selected solely on the basis of location. The criteria (parameters) discussed in this paper are found to be much more relevant and significant to residence selection than the criteria previously discussed by many researchers in different studies. The suggested study is concerned with the interval-valued fuzzy hypersoft environment, which means that sub-parameters are also taken into consideration and may fruitfully fulfill the requirements of the user environment.

## Operational role of selected parameters

1. **Location**: Location has a tremendous influence on the housing market. Housing units vary depending on their surroundings, the type of neighborhood they are located in, and how close they are to centers of work and retail. This is because their locations are set, making them immobile. The location-specific area additionally indicates that a

**Table 7  Weighted fuzzy decision matrix V$_W$.**

| | | | |
|---|---|---|---|
| $\check{\zeta}_{11}$ | $\check{\zeta}_{12}$ | $\cdots$ | $\check{\zeta}_{1n}$ |
| $\check{\zeta}_{21}$ | $\check{\zeta}_{22}$ | $\cdots$ | $\check{\zeta}_{2n}$ |
| $\vdots$ | $\vdots$ | | $\vdots$ |
| $\check{\zeta}_{m1}$ | $\check{\zeta}_{m2}$ | $\cdots$ | $\check{\zeta}_{mn}$ |

**Table 8  Linguistic term for evaluation of parameters.**

| Attribute | Description | Measuring units |
|---|---|---|
| Location | Non-Ideal (not main city), ideal (main city) | – |
| Price | American dollars | million $ |
| Plot size | dimensions/area of plot | square meter |
| Adaptation to weather | Design of house in accordance wit climate | - |
| Covered area | Area covered by infrastructure (building) | part of total area |
| Bed rooms | rooms with at-least 1 window | number |
| Access to main road | Distance from main road/highway | kilometer |
| Bath rooms | including bathrooms attached to bedrooms | number |
| Floor | single story, multi story | number |
| Green surrounding | public park nearby | – |
| Recreation facility | cinema, zoo nearby | – |
| Educational Institution | distance | meter |
| Health Facility | public hospital nearby | meter |
| Security and safety system | minimum time for arrival of emergency services | minutes |
| Market | distance | meter |
| Architecture | Material used | – |

home's environment may have a serious influence on how much it is worth. Residences in town committees and housing schemes have more value than others.

2. **Price**: The cost of a residence is directly affected by a number of factors. Location and other physical facilities boost the price of the residence. Approximating the appropriate manufacturing cost is also necessary as it indicates the proposed worth of the residence.

3. **Plot size**: The shape and size of the plot are important factors as they directly affect the physical appearance and beauty of a residential building. Square-shaped plots with a size of about 300 square meters are preferred over rectangular plots for symmetric designs.

4. **Adaptation to weather**: The basic concept is to make sure that the design of the house is in accordance with the climate. For example, being extremely cold or hot during the winter or summer months, respectively, can affect the air conditioning and heating systems. As well as heat preservation systems like insulation, both are very important.

5. **Covered area**: The ratio of the constructed portion of land *versus* the unconstructed portion varies with location and need. Normally, in metropolitan areas, the constructed portion dominates the unconstructed portion of residences. For a healthy environment, this ratio should be 1 : 1.

6. **Number of bedrooms**: The bedroom serves as the most significant room in the house in many respects. The importance of getting decent sleep is becoming increasingly clear given the pressures that the digital age throws on our time. In addition to enhancing focus and productivity, enough sleep has been linked to improved immune system performance, mental wellness, and even weight loss. One or two bedrooms are enough for a small family, but demand increases with the increase in the number of individuals in the family.

7. **Access to the main road**: The worth of a residence varies with distance from the main road/highway. Residences near main roads and highways have more value than others.

8. **Number of bathrooms**: A pleasant, warm bath or a hot shower in the bathroom may help you relax after a long day of work and provide that little moment of isolation. This transforms the bathroom into a safe zone rather than simply an additional room in the house. The number of bathrooms varies with the density of the population in the building. Normally, at least one bathroom is necessary for each living room (bedroom).

9. **Number of floors**: A building with several stories above the ground is referred to as a ''multi-story building'' in this context. Typically, a structure is considered a multi-story if it comprises more than two levels. Living in a building like that has both benefits and drawbacks.

10. **Green surrounding**: Numerous health advantages, particularly reduced illness and premature death, increased average lifespan, fewer mental health issues, decreased heart disease, improved mental abilities in adolescents and elders, and healthier infants, are linked to green space. A residential building with green surroundings is preferable to a building in or near an industrial zone.

11. **Recreation facility**: Zoos and cinemas are the places where people go for recreation. Zoos are especially popular with children. Zoos also put a great deal of attention on scientific study and wildlife management in addition to entertaining and instructing the general public. It is becoming increasingly popular to give animals more room and recreate their natural environments. The availability of playgrounds and public parks also has certain positive impacts on society. A residential building near such places is indeed a blessing.

12. **Educational Institution**: Students who live near to a school can commute by bike or on foot, breathing in the fresh air while doing their part to preserve the environment. The burden on parents to plan for the school run on a hectic morning is further reduced by easy access to the school. In order to avoid having to go too far in quest of these essential facilities, people hunt for residences close to important amenities like public transit, stores, and schools. The school catchment regions are becoming increasingly important to many home purchasers and renters.

13. **Health Facility**: You can maintain your lifestyle quality by having access to high-quality health-care close by. The ease of routine check-ups, peace of mind during emergencies, and reduced travel time can all be attributed to living close to a hospital.

14. **Security and safety system**: Protecting individuals and their personal property against dangers, such as theft, violence, vandalism, or indeed any hazard that might endanger resident safety, is the goal of residential security. In a nutshell, their responsibility is to

identify risks, look into them, and quickly take appropriate action. To buy a residence, the security plans and safety systems of that locality should be taken into consideration.

15. **Market**: The availability of fresh, regionally produced food and frequently cheap organic food at public markets benefits the overall health of the neighborhood. They also serve as the focal point of the area, promoting pride. Neighborhood markets contain the freshest and healthiest products and normally include locals or farmers selling their wares.

16. **Materials used in architecture**: Building materials signify structural existence. It proves the prevalence of a sense of beauty in a design and, as a result, confirms the stainability of the construction.

## Application

Finding the precise image of society in terms of the indicators needed to choose a residence from the possible choices was the initial stage of this exploratory work. The data was obtained through conversations among residents of the region, comprising estate advertising or marketing personnel or representatives, professionals and personalized specialists, respondents with previous experience in selecting or acquiring a home, and individuals with no previous experience in residence choice. A discussion was undertaken, and the participants were given a survey. They had to grade the relevance of the parameters in house choosing on a scale ranging from one to 5. Participants were asked to specify.

The following example (problem) can now be solved step-by-step by employing this group DM algorithm:

**Step 1** Let us say a real estate agent "A" has a variety of homes $\ddot{\theta} = \{\ddot{\theta}_1, \ddot{\theta}_2, \ddot{\theta}_3, \ddot{\theta}_4, \ddot{\theta}_5\}$, each of which may be described by a set of parameters $\mathbf{X} = \{\mho_1, \mho_2, \mho_3, ..., \mho_{16}\}$ along with the description and measuring units, as elaborated in Table 8. The parameters $\mho_j$ stand for attributes location, price, plot size, adaptation to weather, covered area, number of bedrooms, access to the main road, number of bathrooms, number of floors, green surrounding, recreation facility nearby, educational institution nearby, health facility nearby, security and safety system, market nearby, and material used for construction, respectively, for $j = 1, 2, 3, ..., 16$. The attribute valued sets corresponding to each parameter along with the prescribed value and fuzzified interval value given below and elaborated in Table 9.

$X_1 = \{\text{non-ideal, ideal}\} = \{\mho_{11}, \mho_{12}\}$

$X_2 = \{\text{Less than or equal to 0.1 M\$, greater than 0.1M\$ up to 0.15M\$, greater than 0.15M\$}\} = \{\mho_{21}, \mho_{22}, \mho_{23}\}$

$X_3 = \{\text{less than or equal to 300, greater than 300}\} = \{\mho_{31}, \mho_{32}\}$

$X_4 = \{\text{extremely hot climate (EH), extremely cold climate (EC), EH\&EC}\} = \{\mho_{41}, \mho_{42}, \mho_{43}\}$

$X_5 = \{\text{less than or equal to 50\%, greater than 50\%}\} = \{\mho_{51}, \mho_{52}\}$

$X_6 = \{\text{less than 3, 3 to 5, greater than 5}\} = \{\mho_{61}, \mho_{62}, \mho_{63}\}$

$X_7 = \{\text{main road, less than 2km, greater than 2km}\} = \{\mho_{71}, \mho_{72}, \mho73\}$

$X_8 = \{\text{less than 3, 3 to 4, more than 4}\} = \{\mho_{81}, \mho_{82}, \mho_{83}\}$

$X_9 = \{1, 2, 3, \text{more than 3}\} = \{\mho_{91}, \mho_{92}, \mho_{93}, \mho_{94}\}$

$X_{10} = \{\text{available, not available}\} = \{\mho_{101}, \mho_{102}\}$

$X_{11} = \{\text{available, not available}\} = \{\mho_{111}, \mho_{112}\}$

$X_{12} = \{\text{less than or equal to 300 m, more than 300 m but less than 1 km, more than 1 km}\} = \{\mho_{121}, \mho_{122}, \mho_{123}\}$

$X_{13} = \{\text{less than or equal to 500 m, more than 500 m}\} = \{\mho_{131}, \mho_{132}\}$

$X_{14} = \{\text{less than or equal to 5 min, greater than 5 min}\} = \{\mho_{141}, \mho_{142}\}$

$X_{15} = \{\text{less than or equal to 500 m, greater than 500 m}\} = \{\mho_{151}, \mho_{152}\}$

$X_{16} = \{\text{concrete, wood}\} = \{\mho_{161}, \mho_{162}\}$

By consulting experts some sub-attributes are preferred over others. In $X_1, \mho_{12}$ is preferred over others. In $X_2, \mho_{22}$ is preferred, in $X_3, \mho_{32}$ is preferred, in $X_4, \mho_{43}$ is preferred, in $X_5, \mho_{51}$ and $\mho_{52}$ are given equal preference. In $X_6, \mho_{62}$ is preferred, in $X_7, \mho_{72}$ is preferred, in $X_8, \mho_{82}$ is preferred, in $X_9, \mho_{91}, \mho_{92}$ are given equal preference. In $X_{10}, \mho_{101}$ is preferred, in $X_{11}, \mho_{111}$ is preferred, in $X_{12}, \mho_{122}$ is preferred, in $X_{13}, \mho_{131}$ is preferred, in $X_{14}, \mho_{141}$ is preferred, in $X_{15}, \mho_{151}$ is preferred, and in $X_{16}, \mho_{161}$ and $\mho_{162}$ are given equal preference. Now $\mathfrak{X}^\star = X_1 \times X_2 \times ... X_{16} = \{\mho_1^\star, \mho_2^\star, ..., \mho_8^\star\}$ where each $\mho_i^\star, i = 1, 2, ..., 8$ is a sixteen tuple element. Then consider the following examples: Consider a situation where a real estate agent is approached by three decision-makers $\Delta_1, \Delta_2$ and $\Delta_3$ to purchase a residential building. Prior to making a decision, each decision-maker must take into account his/her own set of criteria. Following that, they can build their interval-valued fuzzy hypersoft sets. Next, we choose a residential building based on the specifications of the sets of decision-makers by using OOPCS as a fuzzy set theory DM approach. Assume that $\Delta_1, \Delta_2, \Delta_3$ create their respective interval-valued fuzzy hypersoft sets $\mathbf{D}_1^\star, \mathbf{D}_2^\star, \mathbf{D}_3^\star$ respectively that are displayed in the form of matrices as follow;

$$
\mathbf{D}_1^\star = 
\begin{array}{c}
\ddot{\theta}_1 \\ \ddot{\theta}_2 \\ \ddot{\theta}_3 \\ \ddot{\theta}_4 \\ \ddot{\theta}_5
\end{array}
\begin{pmatrix}
[0.1,0.3] & [0.2,0.5] & [0.3,0.5] & [0.2,0.6] & [0.4,0.5] & [0.2,0.7] & [0.5,0.7] & [0.4,0.5] \\
[0.4,0.6] & [0.4,0.5] & [0.7,0.8] & [0.3,0.5] & [0.5,0.6] & [0.4,0.7] & [0.3,0.5] & [0.1,0.4] \\
[0.3,0.5] & [0.2,0.6] & [0.4,0.5] & [0.4,0.7] & [0.3,0.6] & [0.4,0.6] & [0.2,0.3] & [0.4,0.6] \\
[0.5,0.7] & [0.2,0.6] & [0.2,0.4] & [0.2,0.6] & [0.3,0.4] & [0.1,0.3] & [0.6,0.7] & [0.5,0.6] \\
[0.4,0.6] & [0.5,0.8] & [0.3,0.7] & [0.6,0.8] & [0.5,0.7] & [0.3,0.5] & [0.4,0.6] & [0.3,0.5]
\end{pmatrix}
$$

with columns $\mho_1^*, \mho_2^*, \mho_3^*, \mho_4^*, \mho_5^*, \mho_6^*, \mho_7^*, \mho_8^*$

$$
\mathbf{D}_2^\star = 
\begin{array}{c}
\ddot{\theta}_1 \\ \ddot{\theta}_2 \\ \ddot{\theta}_3 \\ \ddot{\theta}_4 \\ \ddot{\theta}_5
\end{array}
\begin{pmatrix}
[0.2,0.3] & [0.4,0.6] & [0.6,0.8] & [0.5,0.6] & [0.3,0.7] & [0.6,0.8] & [0.3,0.4] & [0.5,0.8] \\
[0.3,0.5] & [0.2,0.4] & [0.4,0.5] & [0.6,0.8] & [0.3,0.7] & [0.5,0.7] & [0.5,0.7] & [0.3,0.5] \\
[0.5,0.7] & [0.4,0.6] & [0.2,0.3] & [0.3,0.5] & [0.2,0.4] & [0.2,0.6] & [0.4,0.7] & [0.5,0.8] \\
[0.5,0.6] & [0.7,0.9] & [0.2,0.4] & [0.3,0.5] & [0.5,0.6] & [0.3,0.7] & [0.4,0.6] & [0.2,0.4] \\
[0.5,0.6] & [0.3,0.5] & [0.4,0.7] & [0.5,0.6] & [0.6,0.7] & [0.3,0.5] & [0.2,0.3] & [0.6,0.7]
\end{pmatrix}
$$

with columns $\mho_1^*, \mho_2^*, \mho_3^*, \mho_4^*, \mho_5^*, \mho_6^*, \mho_7^*, \mho_8^*$

$$
\mathbf{D}_3^\star = 
\begin{array}{c}
\ddot{\theta}_1 \\ \ddot{\theta}_2 \\ \ddot{\theta}_3 \\ \ddot{\theta}_4 \\ \ddot{\theta}_5
\end{array}
\begin{pmatrix}
[0.1,0.2] & [0.3,0.4] & [0.2,0.3] & [0.4,0.5] & [0.4,0.6] & [0.4,0.5] & [0.2,0.5] & [0.3,0.7] \\
[0.3,0.4] & [0.3,0.6] & [0.2,0.5] & [0.1,0.3] & [0.4,0.5] & [0.2,0.5] & [0.5,0.6] & [0.3,0.6] \\
[0.5,0.6] & [0.3,0.4] & [0.5,0.8] & [0.5,0.6] & [0.5,0.8] & [0.2,0.5] & [0.4,0.7] & [0.2,0.4] \\
[0.2,0.6] & [0.4,0.5] & [0.3,0.6] & [0.6,0.8] & [0.7,0.8] & [0.3,0.4] & [0.3,0.5] & [0.2,0.5] \\
[0.7,0.9] & [0.2,0.3] & [0.4,0.7] & [0.3,0.5] & [0.6,0.7] & [0.4,0.6] & [0.3,0.5] & [0.4,0.8]
\end{pmatrix}
$$

with columns $\mho_1^*, \mho_2^*, \mho_3^*, \mho_4^*, \mho_5^*, \mho_6^*, \mho_7^*, \mho_8^*$

**Step 2** Construction of weighed interval-valued fuzzy parameter matrix MD displayed in Table 10:

**Step 3** Mean difference of weighted interval-valued fuzzy parameter matrix M by $x = \frac{b-a}{2}$ obtained for interval $(a, b)$ is displayed in the Table 11

**Step 4** Construction of weighted vector $\mathbf{W}$ by utilizing Eq. (9), as follow $\varpi_1 = \frac{0.1 + 0.1 + 0.1}{3} = 0.1$, $\varpi_2 = \frac{0.15 + 0.05 + 0.2}{3} = 0.133$, $\varpi_3 = \frac{0.1 + 0.05 + 0.05}{3} = 0.066$, $\varpi_4 = \frac{0.2 + 0.1 + 0.15}{3} = 0.15$, $\varpi_5 = \frac{0.05 + 0.05 + 0.15}{3} = 0.083$, $\varpi_6 = \frac{0.25 + 0.15 + 0.1}{3} = 0.166$, $\varpi_7 = \frac{0.1 + 0.1 + 0.05}{3} = 0.083$, $\varpi_8 = \frac{0.05 + 0.15 + 0.1}{3} = 0.1$ So $\sum_{i=1}^{8} \varpi_i = \varpi_1 + \varpi_2 + \varpi_3 + \varpi_4 + \varpi_5 +$

$\varpi_6 + \varpi_7 + \varpi_8 = 0.881$. Therefore $W_1 = \frac{0.1}{0.881} = 0.11351$, $W_2 = \frac{0.133}{0.881} = 0.15096$, $W_3 = \frac{0.066}{0.881} = 0.07491$, $W_4 = \frac{0.15}{0.881} = 0.17026$, $W_5 = \frac{0.083}{0.881} = 0.09421$, $W_6 = \frac{0.166}{0.881} = 0.18842$, $W_7 = \frac{0.083}{0.881} = 0.09421$, $W_8 = \frac{0.1}{0.881} = 0.11351$. Hence $W = (0.11351, 0.15096, 0.07491, 0.17026, 0.09421, 0.18842, 0.09421, 0.11351)$

**Step 5** Assume that decision makers $\Delta_1, \Delta_2, \Delta_3$ create their respective interval-valued fuzzy hypersoft sets $D_1^\star, D_2^\star$ and $D_3^\star$ that are displayed in Tables 12, 13 and 14 respectively;

**Step 6** Construction of average interval-valued fuzzy hypersoft decision matrix using Eq. (10) displayed in Table 15

**Step 7** Construction of Mean difference of average interval-valued fuzzy parameter matrix $V^\star$ displayed in Table 16 by $x = \frac{b-a}{2}$ for interval $(a, b)$

**Step 8** Weighted fuzzy decision matrix $V^\star$ displayed in Table 17 is constructed utilizing Eq. (11) as follow:

**Step 9** Fuzzy valued $+ive$ ideal solution $(^+IS)$ and $-ive$ ideal solution $(^-IS)$ can be obtained using Eqs. (12) and (13) as follow

$$A^+ = {}^+IS = \begin{Bmatrix} \check{\xi}_1^+ = 0.0132432117, \check{\xi}_2^+ = 0.0176125032, \check{\xi}_3^+ = 0.0124852497, \\ \check{\xi}_4^+ = 0.0227007658, \check{\xi}_5^+ = 0.0125619614, \check{\xi}_6^+ = 0.0282630000, \\ \check{\xi}_7^+ = 0.0109914807, \check{\xi}_8^+ = 0.0151354234 \end{Bmatrix}$$

$$A^- = {}^-IS = \begin{Bmatrix} \check{\xi}_1^- = 0.0075677117, \check{\xi}_2^- = 0.0150960000, \check{\xi}_3^- = 0.0062422503, \\ \check{\xi}_4^- = 0.0141877658, \check{\xi}_5^- = 0.0047105000, \check{\xi}_6^- = 0.0188420000, \\ \check{\xi}_7^- = 0.0078514614, \check{\xi}_8^- = 0.0113510000 \end{Bmatrix}$$

**Step 10** Calculating $SM_i^+$ and $SM_i^-$ for $i = 1, 2, 3$ from Eqs. (5) and (6)

$$SM_1^+ = \sqrt{\begin{Bmatrix} (0.0132432117 - 0.0075677117)^2 + (0.0176125032 - 0.0150960000)^2 + \\ (0.0124852497 - 0.0062422503)^2 + (0.0227007658 - 0.0170260000)^2 + \\ (0.0125619614 - 0.0109914807)^2 + (0.0282630000 - 0.0251239228)^2 + \\ (0.0109914807 - 0.0094210000)^2 + (0.0151354234 - 0.0151354234)^2 \end{Bmatrix}}$$

$$= \sqrt{\begin{Bmatrix} (0.0056755000)^2 + (0.0025165032)^2 + \\ (0.0062429994)^2 + (0.0056747658)^2 + \\ (0.0015704807)^2 + (0.0031390772)^2 + \\ (0.0015704807)^2 + (0)^2 \end{Bmatrix}}$$

$$= \sqrt{\begin{Bmatrix} 0.00003221130 + 0.00000633279 + \\ 0.00003897504 + 0.00003220297 + \\ 0.00000246641 + 0.00000985381 + \\ 0.00000246641 + 0 \end{Bmatrix}}$$

$$= \sqrt{0.00012450873} = 0.011158348$$

Similarly $SM_1^-, SM_2^+, SM_2^-, SM_3^+, SM_3^-, SM_4^+, SM_4^-, SM_5^+, SM_5^-$ can be calculated: $SM_1^- = 0.01018597849$, $SM_2^+ = 0.01101842605$, $SM_2^- = 0.00935871757$, $SM_3^+ = 0.00943778278$, $SM_3^- = 0.01348368274$, $SM_4^+ = 0.01180231684$, $SM_4^- = 0.01127467957$, $SM_5^+ = 0.01532198987$, $SM_5^- = 0.00697154082$

**Step 11** Relative closeness of alternatives to the ideal solution can be calculated as follow: $RC_1^+ = \frac{SM_1^-}{SM_1^- + SM_1^+} = \frac{0.01018597849}{0.01018597849 + 0.011158348} = \frac{0.01018597849}{0.02134432649} = 0.4772218273$ Hence $RC_1^+ = 0.4772218273$ Similarly $RC_2^+ = 0.45927524213$, $RC_3^+ = 0.5882556998$, $RC_4^+ = 0.48856789548$, $RC_5^+ = 0.31271586887$

**Step 12** Ranking the preference order is $\ddot{\theta}_5 \leq \ddot{\theta}_2 \leq \ddot{\theta}_1 \leq \ddot{\theta}_4 \leq \ddot{\theta}_3$ As benefit parameters are preferred over cost parameters, so $\ddot{\theta}_3$ is selected. The pictorial form of ranking is shown in Fig. 4.

**Table 9 Linguistic term for evaluation of parameters.**

| Attribute | Sub-attribute | Prescribed value | Fuzzy value interval |
|---|---|---|---|
| Location | non-ideal, ideal | 0,1 | $[0, 0.50], [0.51, 1]$ |
| Price(p) | Less than or equal to 0.1 M\$, greater than 0.1M\$ up to 0.15M\$, greater than 0.15M\$ | $0 \leq p \leq 0.1, 0.1 < p \leq 0.15, p > 0.15$ | $[0, 0.33], [0.34, 0.66], [0.67, 1]$ |
| Plot size(s) | less than or equal to 300 m$^2$, greater than 300 m$^2$ | $0 \leq s \leq 300, s > 300$ | $[0, 0.50], [0.51, 1]$ |
| Adaptation to weather | extremely hot climate (EH), extremely cold climate (EC), EH&EC | 1,2,3 | $[0, 0.33], [0.34, 0.66], [0.67, 1]$ |
| Covered area | less than or equal to 50%, greater than 50% | 1,2 | $[0, 0.50], [0.51, 1]$ |
| Bed rooms | less than 3, 3 to 5, greater than 5 | 2,5,6 | $[0, 0.33], [0.34, 0.66], [0.67, 1]$ |
| Access to main road | main road, less than 2km, greater than 2km | 0,1,2 | $[0, 0.33], [0.34, 0.67], [0.67, 1]$ |
| Bath rooms | less than 3, 3 to 4, more than 4 | 2,4,5 | $[0, 0.33], [0.34, 0.66], [0.67, 1]$ |
| Floor | 1, 2, 3, more than 3 | 0,1,2,3 | $[0, 0.25], [0.26, 0.50], [0.51, 0.75], [0.76, 1]$ |
| Green surrounding | available / not available | 0,1 | $[0, 0.50], [0.51, 1]$ |
| Recreation facility | available / not available | 0,1 | $[0, 0.50], [0.51, 1]$ |
| Educational Institution | less than or equal to 300 m, more than 300 m but less than 1 km, more than 1 km | 0,1,2 | $[0, 0.33], [0.34, 0.66], [0.67, 1]$ |
| Health Facility | less than or equal to 500 m, more than 500 m | 1,2 | $[0, 0.50], [0.51, 1]$ |
| Security and safety system | less than or equal to 5 min, greater than 5 min | 1,2 | $[0, 0.50], [0.51, 1]$ |
| Market | less than or equal to 500 m, greater than 500 m | 1,2 | $[0, 0.50], [0.51, 1]$ |
| Architecture | concrete, wood | 1,2 | $[0, 0.50], [0.51, 1]$ |

**Table 10 Weighted interval-valued fuzzy parameter matrix MD.**

| MD | $\mho_1^\star$ | $\mho_2^\star$ | $\mho_3^\star$ | $\mho_4^\star$ | $\mho_5^\star$ | $\mho_6^\star$ | $\mho_7^\star$ | $\mho_8^\star$ |
|---|---|---|---|---|---|---|---|---|
| $\Delta_1$ | $[0.2, 0.4]$ | $[0.4, 0.7]$ | $[0.2, 0.4]$ | $[0.3, 0.7]$ | $[0.6, 0.7]$ | $[0.1, 0.6]$ | $[0.4, 0.6]$ | $[0.7, 0.8]$ |
| $\Delta_2$ | $[0.2, 0.4]$ | $[0.3, 0.4]$ | $[0.6, 0.7]$ | $[0.5, 0.7]$ | $[0.6, 0.7]$ | $[0.6, 0.9]$ | $[0.6, 0.8]$ | $[0.2, 0.5]$ |
| $\Delta_3$ | $[0.4, 0.6]$ | $[0.3, 0.7]$ | $[0.6, 0.7]$ | $[0.1, 0.4]$ | $[0.2, 0.5]$ | $[0.5, 0.7]$ | $[0.5, 0.6]$ | $[0.6, 0.8]$ |

## DISCUSSION AND COMPARATIVE ANALYSIS

In this research article, a new DM strategy called "OOPCS" has been introduced for interval-valued data in addition to the partitioning of attributes into non-overlapping attribute-valued sets. Many researchers have studied the TOPSIS technique in the literature, however, the majority of them concentrate on attributes instead of sub-attribute values in the form of sets. Many real-world situations necessarily require the partitioning of attributes into distinct attribute value sets. Ignoring the value of a sub-attribute can result in confusion and have a negative influence on the decision. The inclusion of interval settings allows this strategy, "OOPCS" to counter all situations having interval-form data, which increases the efficiency of the proposed model.

Many academics have investigated how to choose the best option out of the options for an unclear MADM problem. Now, we contrast our calculated outcomes and ranking with those of other studies. These scores are first translated to pertinent fuzzy values by dividing each scoring value by the greatest score, even if they have been computed in terms

**Table 11 Mean difference of weighted interval-valued fuzzy parameter matrix M.**

| M | $\mho_1^\star$ | $\mho_2^\star$ | $\mho_3^\star$ | $\mho_4^\star$ | $\mho_5^\star$ | $\mho_6^\star$ | $\mho_7^\star$ | $\mho_8^\star$ |
|---|---|---|---|---|---|---|---|---|
| $\Delta_1$ | 0.1 | 0.15 | 0.1 | 0.2 | 0.05 | 0.25 | 0.1 | 0.05 |
| $\Delta_2$ | 0.1 | 0.05 | 0.05 | 0.1 | 0.05 | 0.15 | 0.1 | 0.15 |
| $\Delta_3$ | 0.1 | 0.2 | 0.05 | 0.15 | 0.15 | 0.1 | 0.05 | 0.1 |

**Table 12 Interval-valued fuzzy hypersoft set $D_1^\star$.**

| $D_1^\star$ | $\mho_1^\star$ | $\mho_2^\star$ | $\mho_3^\star$ | $\mho_4^\star$ | $\mho_5^\star$ | $\mho_6^\star$ | $\mho_7^\star$ | $\mho_8^\star$ |
|---|---|---|---|---|---|---|---|---|
| $\ddot{\theta}_1$ | [0.1, 0.3] | [0.2, 0.5] | [0.3, 0.5] | [0.2, 0.6] | [0.4, 0.5] | [0.2, 0.7] | [0.5, 0.7] | [0.4, 0.5] |
| $\ddot{\theta}_2$ | [0.4, 0.6] | [0.4, 0.5] | [0.7, 0.8] | [0.3, 0.5] | [0.5, 0.6] | [0.4, 0.7] | [0.3, 0.5] | [0.1, 0.4] |
| $\ddot{\theta}_3$ | [0.3, 0.5] | [0.2, 0.6] | [0.4, 0.5] | [0.4, 0.7] | [0.3, 0.6] | [0.4, 0.6] | [0.2, 0.3] | [0.4, 0.6] |
| $\ddot{\theta}_4$ | [0.5, 0.7] | [0.2, 0.6] | [0.2, 0.4] | [0.2, 0.6] | [0.3, 0.4] | [0.1, 0.3] | [0.6, 0.7] | [0.5, 0.6] |
| $\ddot{\theta}_5$ | [0.4, 0.6] | [0.5, 0.8] | [0.3, 0.7] | [0.6, 0.8] | [0.5, 0.7] | [0.3, 0.5] | [0.4, 0.6] | [0.3, 0.5] |

**Table 13 Interval-valued fuzzy hypersoft set $D_2^\star$.**

| $D_2^\star$ | $\mho_1^\star$ | $\mho_2^\star$ | $\mho_3^\star$ | $\mho_4^\star$ | $\mho_5^\star$ | $\mho_6^\star$ | $\mho_7^\star$ | $\mho_8^\star$ |
|---|---|---|---|---|---|---|---|---|
| $\ddot{\theta}_1$ | [0.2, 0.3] | [0.4, 0.6] | [0.6, 0.8] | [0.5, 0.6] | [0.3, 0.7] | [0.6, 0.8] | [0.3, 0.4] | [0.5, 0.8] |
| $\ddot{\theta}_2$ | [0.3, 0.5] | [0.2, 0.4] | [0.4, 0.5] | [0.6, 0.8] | [0.3, 0.7] | [0.5, 0.7] | [0.5, 0.7] | [0.3, 0.5] |
| $\ddot{\theta}_3$ | [0.5, 0.7] | [0.4, 0.6] | [0.2, 0.3] | [0.3, 0.5] | [0.2, 0.4] | [0.2, 0.6] | [0.4, 0.7] | [0.5, 0.8] |
| $\ddot{\theta}_4$ | [0.5, 0.6] | [0.7, 0.9] | [0.2, 0.4] | [0.3, 0.5] | [0.5, 0.6] | [0.3, 0.7] | [0.4, 0.6] | [0.2, 0.4] |
| $\ddot{\theta}_5$ | [0.5, 0.6] | [0.3, 0.5] | [0.4, 0.7] | [0.5, 0.6] | [0.6, 0.7] | [0.3, 0.5] | [0.2, 0.3] | [0.6, 0.7] |

of discrete values that are compatible with our framework. The sensitivity analysis of score values and comparison are presented in Tables 18 and 19. Because it is an established principle in numerical mathematics that "the smaller the values, the more reliable and precise the results are considered," the tables clearly show that the score values obtained through the algorithm we recommend are more consistent and reliable. Next, we evaluate the benefits of our proposed model by comparing it with certain pertinent current models while taking into account several significant evaluation factors, such as IV-settings, fuzzy membership grades, SS = soft setting, and HS = hypersoft setting. This comparison, also referred to as a structural comparison, evaluates the model's adaptability. From Table 19, It is obvious that the majority of the current models are unique examples of our suggested model, demonstrating the adaptability of our concept.

The positive aspects of the suggested study are as follows:

1. Interval values are used to collect data instead of fuzzy values because they are more dependable in uncertain natural environments.
2. The model is more comprehensive and proficient as it relies on sub-parameters instead of parameters.
3. Sixteen possible parameters and their respective sub-parametric valued sets are considered for the selection of a residential building, which has enhanced the scope of the DM problem.

**Table 14  Interval-valued fuzzy hypersoft set $D_3^\star$.**

| $D_3^\star$ | $\mho_1^\star$ | $\mho_2^\star$ | $\mho_3^\star$ | $\mho_4^\star$ | $\mho_5^\star$ | $\mho_6^\star$ | $\mho_7^\star$ | $\mho_8^\star$ |
|---|---|---|---|---|---|---|---|---|
| $\ddot\theta_1$ | [0.1, 0.2] | [0.3, 0.4] | [0.2, 0.3] | [0.4, 0.5] | [0.4, 0.6] | [0.4, 0.5] | [0.2, 0.5] | [0.3, 0.7] |
| $\ddot\theta_2$ | [0.3, 0.4] | [0.3, 0.6] | [0.2, 0.5] | [0.1, 0.3] | [0.4, 0.5] | [0.2, 0.5] | [0.5, 0.6] | [0.3, 0.6] |
| $\ddot\theta_3$ | [0.5, 0.6] | [0.3, 0.4] | [0.5, 0.8] | [0.5, 0.6] | [0.5, 0.8] | [0.2, 0.5] | [0.4, 0.7] | [0.2, 0.4] |
| $\ddot\theta_4$ | [0.2, 0.6] | [0.4, 0.5] | [0.3, 0.6] | [0.6, 0.8] | [0.7, 0.8] | [0.3, 0.4] | [0.3, 0.5] | [0.2, 0.5] |
| $\ddot\theta_5$ | [0.7, 0.9] | [0.2, 0.3] | [0.4, 0.7] | [0.3, 0.5] | [0.6, 0.7] | [0.4, 0.6] | [0.3, 0.5] | [0.4, 0.8] |

**Table 15  Average interval-valued fuzzy hypersoft set $V^\star$.**

| $V^\star$ | $\mho_1^\star$ | $\mho_2^\star$ | $\mho_3^\star$ | $\mho_4^\star$ |
|---|---|---|---|---|
| $\ddot\theta_1$ | [0.13333, 0.26667] | [0.30000, 0.50000] | [0.36667, 0.53333] | [0.36667, 0.56667] |
| $\ddot\theta_2$ | [0.33333, 0.50000] | [0.30000, 0.50000] | [0.43333, 0.60000] | [0.33333, 0.53333] |
| $\ddot\theta_3$ | [0.43333, 0.60000] | [0.30000, 0.53333] | [0.36667, 0.53333] | [0.40000, 0.60000] |
| $\ddot\theta_4$ | [0.40000, 0.63333] | [0.43333, 0.66667] | [0.23333, 0.46667] | [0.36667, 0.63333] |
| $\ddot\theta_5$ | [0.53333, 0.70000] | [0.33333, 0.53333] | [0.36667, 0.70000] | [0.46667, 0.63333] |
| | $\mho_5^\star$ | $\mho_6^\star$ | $\mho_7^\star$ | $\mho_8^\star$ |
| $\ddot\theta_1$ | [0.36667, 0.60000] | [0.40000, 0.66667] | [0.33333, 0.53333] | [0.40000, 0.66667] |
| $\ddot\theta_2$ | [0.40000, 0.60000] | [0.36667, 0.63333] | [0.43333, 0.60000] | [0.23333, 0.50000] |
| $\ddot\theta_3$ | [0.33333, 0.60000] | [0.26667, 0.56667] | [0.33333, 0.56667] | [0.36667, 0.60000] |
| $\ddot\theta_4$ | [0.50000, 0.60000] | [0.23333, 0.46667] | [0.43333, 0.60000] | [0.30000, 0.50000] |
| $\ddot\theta_5$ | [0.56667, 0.70000] | [0.33333, 0.53333] | [0.30000, 0.46667] | [0.43333, 0.66667] |

**Table 16  Mean difference of average interval-valued fuzzy parameter matrix $V^\star$.**

| $V^\star$ | $\mho_1^\star$ | $\mho_2^\star$ | $\mho_3^\star$ | $\mho_4^\star$ | $\mho_5^\star$ | $\mho_6^\star$ | $\mho_7^\star$ | $\mho_8^\star$ |
|---|---|---|---|---|---|---|---|---|
| $\ddot\theta_1$ | 0.06667 | 0.10000 | 0.08333 | 0.10000 | 0.11667 | 0.13334 | 0.10000 | 0.13334 |
| $\ddot\theta_2$ | 0.08334 | 0.10000 | 0.08334 | 0.10000 | 0.10000 | 0.13333 | 0.08334 | 0.13334 |
| $\ddot\theta_3$ | 0.08334 | 0.11667 | 0.08333 | 0.10000 | 0.13334 | 0.15000 | 0.11667 | 0.11667 |
| $\ddot\theta_4$ | 0.11667 | 0.11667 | 0.11667 | 0.13333 | 0.05000 | 0.11667 | 0.08334 | 0.10000 |
| $\ddot\theta_5$ | 0.08334 | 0.10000 | 0.16667 | 0.08333 | 0.06667 | 0.10000 | 0.08334 | 0.11667 |

4. The proposed model is more generalized and advanced as it utilizes interval data under the cover of a fuzzy hypersoft set.

5. The current model focuses on the primary investigation of characteristics under HS-settings. It pursues the choice-making best, delicate and additional stable.

6. The suggested model has all the features and characteristics of existing models like F-set, S-set, FS-set, IVF-set, IVFS-set, HS-set and FHS-set.

The comparison analysis of the proposed model to existing soft set-like models is demonstrated in Table 20. The proposed technique is assessed for this study's readers in terms of its logical consistency and computational simplicity. In comparison to previously created methodologies, it is considered that the provided methodology better implements the chosen important criteria. It is proven that this strategy is superior to others as displayed in Table 20. In this table, some prominent characteristics of existing models have been

**Table 17  Weighted fuzzy decision matrix V⋆.**

| V⋆ | $\mho_1^\star$ | $\mho_2^\star$ | $\mho_3^\star$ | $\mho_4^\star$ |
|---|---|---|---|---|
| $\ddot{\theta}_1$ | 0.0075677117 | 0.0150960000 | 0.0062422503 | 0.0170260000 |
| $\ddot{\theta}_2$ | 0.0094599234 | 0.0150960000 | 0.0062429994 | 0.0170260000 |
| $\ddot{\theta}_3$ | 0.0094599234 | 0.0176125032 | 0.0062422503 | 0.0170260000 |
| $\ddot{\theta}_4$ | 0.0132432117 | 0.0176125032 | 0.0087397497 | 0.0227007658 |
| $\ddot{\theta}_5$ | 0.0094599234 | 0.0150960000 | 0.0124852497 | 0.0141877658 |
| | $\mho_5^\star$ | $\mho_6^\star$ | $\mho_7^\star$ | $\mho_8^\star$ |
| $\ddot{\theta}_1$ | 0.0109914807 | 0.0251239228 | 0.0094210000 | 0.0151354234 |
| $\ddot{\theta}_2$ | 0.0094210000 | 0.0251220386 | 0.0078514614 | 0.0151354234 |
| $\ddot{\theta}_3$ | 0.0125619614 | 0.0282630000 | 0.0109914807 | 0.0132432117 |
| $\ddot{\theta}_4$ | 0.0047105000 | 0.0219829614 | 0.0078514614 | 0.0113510000 |
| $\ddot{\theta}_5$ | 0.0062809807 | 0.0188420000 | 0.0078514614 | 0.0132432117 |

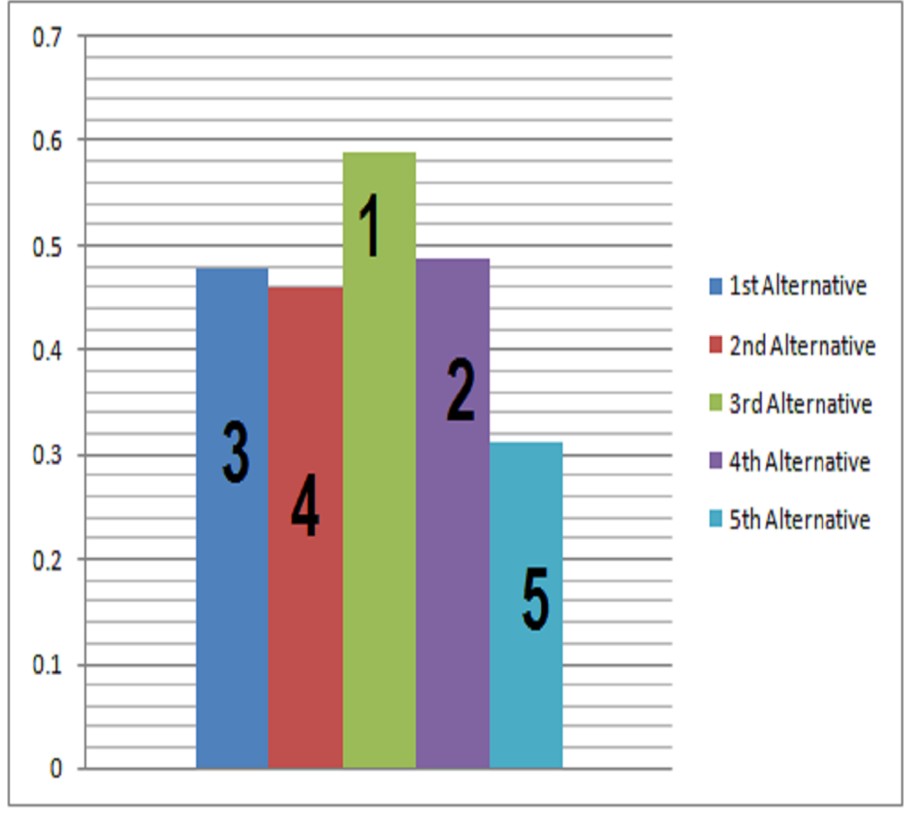

**Figure 4  Decision making based on OOPCS.**

compared with the IVFHS—set. These characteristics include the interval nature of data, the membership function, the single argument approximation operator SAAO, and the multi-argument approximation operator MAAO. From Table 20, It is also obvious that our suggested model is more universal than the models mentioned before.

**Table 18 Sensitivity analysis with ranking.**

| Pythagorean means | Arithmetic mean | Geometric mean | Harmonic mean |
|---|---|---|---|
| $\ddot{\theta}_1$ | 0.01332547361 | 0.00000017492 | 0.01110627274 |
| $\ddot{\theta}_2$ | 0.01316935577 | 0.00000016530 | 0.01095513048 |
| $\ddot{\theta}_3$ | 0.01442504134 | 0.00000024201 | 0.01206824759 |
| $\ddot{\theta}_4$ | 0.01352401915 | 0.00000016532 | 0.01047115465 |
| $\ddot{\theta}_5$ | 0.02282323791 | 0.00000044634 | 0.01245038649 |

**Table 19 Comparison of the anticipated study based on computations.**

| Literature | Fuzzy valued scores | Ranking |
|---|---|---|
| *Tripathy, Sooraj & Mohanty (2017)* | 1.000, 0.941, 0.705, 0.352, 0.529 | $\theta_1 > \theta_2 > \theta_3 > \theta_4 > \theta_5$. |
| Suggested approach | 0.588, 0.489, 0.477, 0.459, 0.313 | $\theta_3 > \theta_4 > \theta_1 > \theta_2 > \theta_5$ |

**Table 20 Comparative study.**

| Author | Structure | IV-setting | Membership Function | SS | HS |
|---|---|---|---|---|---|
| *Zadeh (1965)* | F-set | × | ✓ | × | × |
| *Gorzałczany (1987)* | IVF-set | ✓ | ✓ | × | × |
| *Molodtsov ((1999)* | S-set | × | × | ✓ | × |
| *Maji, Biswas & Roy (2003)* | FS-set | × | ✓ | ✓ | × |
| *Yang et al. (2009)* | IVFS-set | ✓ | ✓ | ✓ | × |
| *Smarandache (2018)* | HS-set | × | × | ✓ | ✓ |
| Proposed Model | Proposed structure | ✓ | ✓ | ✓ | ✓ |

## CONCLUSION

In this article, we have given a technique for group DM utilizing OOPCS in an IVFHS-set environment. Finally, we gave an illustration showing how this approach could be successfully used. It can be used to solve DM issues in a variety of sectors where there is ambiguity. In order to tackle the associated challenges, the technique should be more thorough in the future, and many cases may be suggested for testing in further research. In this research, the ranking of alternatives is done for a real-estate DM problem based on IVFHS setting. In this model, the influence of attributes on DM has been enhanced by taking their respective values from separate attribute value sets. A real state agent is approached by three decision-makers $\Delta_1, \Delta_2, \Delta_3$ to purchase a residential building. The decision-makers have fixed their own set of criteria by considering attributes and sub-attributes. A weighted vector has been constructed based on the interval weights given by the decision maker to each criterion (element of the IVFHS-set) $\Delta_1, \Delta_2$ and $\Delta_3$ constructed their respective IVFHS-sets. This model employs intervals with upper and lower bounds as well as fuzzy values to account for the ambiguous and uncertain character of the data. The HS can be used in this structure to store data in intervals with fuzzy values. There are some limitations in the proposed study to deal with situations like: the situations dealing with periodic nature

of data in a complex plane; the situations involving data, not in the form of intervals and the situations dealing with rough types of data. The research covers almost the entire spectrum of artificial intelligence and soft computing. Many structures, like the intuitionistic F-set, the neutrosophic set, the Pythagorean F-set, the picture F-set, and the refined F-set, can be hybridized with the hypersoft set to form new structures with their application in DM in the coming future. In this study, the decisive comments of decision makers are hypothetical in nature; however, this framework can be applied to any case study with a real data set for such decisive comments. Furthermore, in the IVFHS-set, ordinary product of disjoint subclasses of sub attributes is considered for getting a multi argument domain; if it is considered a cartesian product, then this mathematical framework can easily be extended to develop algebraic structures, topological structures, and convex optimization related problems.

### Funding
The authors received no funding for this work.

### Competing Interests
The authors declare there are no competing interests.

### Author Contributions
- Muhammad Arshad conceived and designed the experiments, performed the experiments, analyzed the data, performed the computation work, prepared figures and/or tables, authored or reviewed drafts of the article, and approved the final draft.
- Muhammad Saeed performed the experiments, analyzed the data, performed the computation work, authored or reviewed drafts of the article, and approved the final draft.
- Atiqe Ur Rahman conceived and designed the experiments, performed the experiments, analyzed the data, performed the computation work, authored or reviewed drafts of the article, and approved the final draft.
- Mazin Abed Mohammed performed the experiments, authored or reviewed drafts of the article, and approved the final draft.
- Karrar Hameed Abdulkareem performed the experiments, authored or reviewed drafts of the article, and approved the final draft.
- Abed Saif Alghawli performed the experiments, authored or reviewed drafts of the article, and approved the final draft.
- Mohammed AA Al-qaness performed the experiments, authored or reviewed drafts of the article, and approved the final draft.

### Data Availability
    The raw data is available in the Supplemental Files.

## Supplemental Information

Supplemental information for this article can be found online at http://dx.doi.org/10.7717/peerj-cs.1423#supplemental-information.

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
