# Peer review of "A robust algorithmic cum integrated approach of interval-valued fuzzy hypersoft set and OOPCS for real estate pursuit"

_PeerJ Computer Science, doi:10.7717/peerj-cs.1423_

## Round 0.1 · original submission · Minor Revisions

Please revise your paper according to the reviewer's comments.
Thanks.

Reviewer 1 ·

Basic reporting

Dear Editor,

I have completed the review of this research work carefully.

My report is in the attached file.

My decision is accepted with minor revision.

Regards

Experimental design

Please see my report is in the attached file.

Validity of the findings

Please see my report is in the attached file.

Additional comments

Please see my report is in the attached file.

Annotated reviews are not available for download in order to protect the identity of reviewers who chose to remain anonymous.

Reviewer 2 ·

Basic reporting

*The use of English is clear, understandable, and sufficient. However, the symbol density and differences used in the study make it difficult to understand the study. Abbreviations used in the study should be explained. For example Definition 1.1. I think the definition of fuzzy set concept has been given. But it is not clear what F-set is. A similar situation exists in other definitions.It should be revised in this respect.
*Literature knowledge about hypersoft sets and fuzzy hypersoft sets should be expanded. For example, the following study and similar studies should be examined and included in the literature.
Fuzzy hypersoft sets and its application to decision-making." Theory and application of hypersoft set 50 (2021)
*Flowcharts, tables and figures in the study facilitate the understanding of the presented theory.
*In this study, a new decision making method has been developed and presented in detail.
*The presented concepts are examined in detail and the method is explained in detail.

Experimental design

Hypersoft sets is a topical issue that researchers have been working on recently. In this study, the authors enriched the structure in terms of decision making and brought a different perspective. The study is sufficient and appropriate in terms of Experimental design.

Validity of the findings

In this article, a new method called OOPCS is developed in a fuzzy interval-valued hyper-soft environment and a group decision making technique is given. An example is presented to show how this approach can be used. I think that the concepts and method presented in the study are valid and important in terms of studies that can be done on this subject.

Reviewer 3 ·

Basic reporting

In what follows, I mention some revisions that are necessary to improve the manuscript:

1. Review paper’s language, there are some grammatical errors and typos.
2. Please, show the motivation of this study and how it can be applied to address other issues.
3. The literture review is weak, it should be improved by mentioning the recent contributions such as (i) Mathematical analysis of generative adversarial networks based on complex picture fuzzy soft information; (ii) New generalization of fuzzy soft sets: $(a,b)$-Fuzzy soft sets; (iii) Generalized frame for orthopair fuzzy sets: $(m,n)$-Fuzzy sets and their applications to multi-criteria decision-making methods; (iv) (2,1)-Fuzzy sets: properties, weighted aggregated operators and their applications to multi-criteria decision-making methods .
4. Recheck the values given in Table 10 and 11.
5. Explain novelty of the proposed technique.
6. Add a conclusion section including future directions,
Investigate how the current concepts can be applied in the view of other hybridizations?

Experimental design

Please, see my basic comments

Validity of the findings

Please, see my basic comments

Additional comments

Please, see my basic comments

---

## Round 0.2 · accepted · Accept

The revised version is suitable for publication.
Thank you

Reviewer 1 ·

Basic reporting

no comment

Experimental design

no comment

Validity of the findings

no comment

Additional comments

Dear Editor, and Author

I would like to inform you that the authors have completed all reviewers' comments one by one. Therefore, I recommend accepting this manuscript in "PeerJ Computer Science".

Best,

Reviewer 2 ·

Basic reporting

Suggested corrections have been made. I recommend that the study be published in its final form.

Experimental design

Suggested corrections have been made. I recommend that the study be published in its final form.

Validity of the findings

Suggested corrections have been made. I recommend that the study be published in its final form.

Reviewer 3 ·

Basic reporting

I examined the revised version. The corrections according to my remarks were made by the authors. So, I recommend that the paper may be accepted for publication.

Experimental design

No more comments, all the comments are addressed satisfactorily.

Validity of the findings

The obtained results are correct.

Additional comments

No more comments, all the comments are addressed satisfactorily.